# Effects of Residual Composition and Distribution on the Structural Characteristics of the Protein

**DOI:** 10.3390/ijms232214263

**Published:** 2022-11-17

**Authors:** Qiaoling Song, Zhenan Wu, Chenghao Jin, Zhichao Yu, Peng Xu, Zhouting Jiang

**Affiliations:** Department of Applied Physics, China Jiliang University, Hangzhou 310018, China

**Keywords:** HNP model, molecular dynamics simulation, hydrophobicity

## Abstract

The effect of ratio and consecutive number of hydrophobic residues in the repeating unit of protein chains was investigated by MD simulation. The modified off-lattice HNP model was applied in this study. The protein chains constituted by different HNP ratios or different numbers of consecutively hydrophobic residues with the same chain length were simulated under a broad temperature range. We concluded that the proteins with higher ratio or larger number of sequentially hydrophobic residues present more orientated and compact structure under a certain low temperature. It is attributed to the lower non-bonded potential energy between H-H residual pairs, especially more hydrophobic residues in a procession among the protein chain. Considering the microscopic structure of the protein, more residue contacts are achieved with the proteins with higher ratios and sequential H residues under the low temperature. Meanwhile, with the ratio and consecutive number of H residues increasing, the distribution of stem length showed a transition from exponential decline to unimodal and even multiple peaks, indicating the specific ordered structure formed. These results provide an insight into 3D structural properties of proteins from their residue sequences, which has a primary structure at molecular level and, ultimately, a practical possibility of applying in biotechnological applications.

## 1. Introduction

Proteins, as compact polymers, show various structural diversity since they have numerous internal degrees. The biological function of a protein strongly depends on its unique structure determined by complicated interactions among residues. The protein chain conformation, which leads to multiple mechanical and thermodynamical properties, even to biological functions, is extensively interesting for industrialists and scientists. The structure formation of polymer chains, such as polymer crystallization, protein folding, and self-assembly of block copolymers, has recently become the focus of attention in chemistry, physics, biology, and material science [1,2,3]. From the viewpoint of polymer statistics, every polymer chain may occasionally return to itself. That is, some points of the free chain trajectory may come within a short reach of one and another, forming a closed loop, also named contacts. The folding of protein chains into compact, unique three-dimensional structures is guided and stabilized by intramolecular interactions between the constituent amino acid residues along the chain [4]. According to Anfinsen’s denaturation and renaturation experiments, it is known that the three-dimensional conformation of a protein is directly determined by its primary structure, the amino acid sequence, according to the global minimum on free energy landscape of a protein [5]. So, in the case of proteins, the residue–residue contacts, which are mainly induced by van der Waals interactions, play an important role in the protein folding process. The residue contacts are defined as two heavy atoms spatially close but far away in amino acid sequences. Residue contact has been proposed to describe the effect of long-range interactions, such as hydrogen bond, hydrophobic–hydrophobic, aromatic–aromatic, or aromatic–polar interactions on the structural level of the protein [6,7]. However, the determination of the three-dimensional structure of a protein molecule from its amino acid sequence remains difficult because the energy that stabilizes the folded conformation is poorly understood and the number of possible structures is extensive [8]. As each protein has a unique and exact amino acid sequence, it is particularly important to study the effect of amino acids sequence on protein structure.

Computational simulation is a powerful method to discover the detailed mechanism of structural information at an atomistic level. Currently, fruitful research has been carried out in the field of structural formation of the macromolecular system by computer simulation to overcome experimental difficulties [9,10,11]. Despite the rapid progress of computer power and the efforts made by many researchers, numerous basic questions, such as the nature of the interactions involved in protein folding, the thermodynamic states involved as a protein folds, and the structural properties of each thermodynamic state, remain unanswered [12]. Molecular dynamics (MD) simulations at atomic resolution could, in principle, monitor molecular interactions with a high accuracy. However, the simulation on realistic all-atom representations of a protein with the numerous freedom degrees limits the running time because of high computational costs. To solve this problem, the coarse-grained model is applied to represent proteins, which preserves the biological information of the protein and saves computational resources. In the coarse-grained model, each protein is viewed as a chain of beads, each bead representing an amino acid residue and each amino acid residue is replaced by a corresponding $C^{\alpha }$ atom [13]. One of the widely used simplified lattice models is the HP model. Two kinds of beads named H (hydrophobic residues) and P (polar residues) construct a protein chain [14]. The hydrophobic energies are attractive in a coarse-grained model, which means H-H contacts play a major role in the protein-structured process. Later, protein engineering experiments suggest that instead of two, three kinds of residues can be effectively substituted for the twenty amino acids [15]. Miyazawa–Jernigan (MJ) modified the interaction matrix based on this model [16]. According to the MJ matrix, the standard HP lattice model was improved by the off-lattice model with three kinds of residues, namely hydrophobic residues (H), neutral residues (N), and hydrophilic residues (P) [17]. The nonlocal interaction term depends on the type of residues. The interaction between H-H residues has a deep primary minimum that gives rise to defining the ordered structure of a protein. The other residue pairs interact repulsively with different strengths. In our model, the backbone of the protein was constituted by simplified residues in a continuous space instead of a cubic lattice. While lacking the chemical detail of all-atom representations, the simplified models are able to characterize both kinetics and thermodynamics during the protein folding process [18,19,20].

In this work, we apply MD simulations on protein chains with different chain lengths, residue composition, and repeat units. The aim of this study is to investigate the effect of hydrophobic properties of residues on the structural features and conformational transition of proteins. The rest of this paper is organized as follows. The simulation results regarding the effects of the chain length, ratio of hydrophobic residues, and repeat units are discussed in Section 2. The methodology, including the construction of the protein chain model and molecular dynamics simulation details, are presented in Section 3, and the paper is concluded in Section 4.

## 2. Results and Discussion

### 2.1. Effect of the Chain Length

#### 2.1.1. Energy

The chain length dependence of potential energy of each bond in a protein chain with specific H3N1P1 as repeating segment is investigated. According to the simulation model, the total bonded energies of each protein are summed of all the bonds. Meanwhile, the non-bonded energy of the protein sample is summed of all residue pairs. To compare the change of potential energy with the temperature rather than the chain length, the four potential energies of the whole protein chain were divided by the number of bonds, the number of bond angles, or the number of dihedral angles of each protein chain with different chain lengths. Figure 1a,b plot the simulation results of bonded potential energy of each bond versus the reduced temperature $T^{*}$ with chain length N=50, 120, 240, and 300 residues, respectively. As shown in Figure 1a, the change of bond-stretching/bending energy per bond as the function of temperature is independent of chain length. Both energies are almost linearly increasing with the temperature increases, which is approved by the theorem of energy equipartition, i.e., the average energy associated with each individual degree of freedom (bond length/angle) is linearly proportional to the system temperature. The bond-torsional energy per bond of proteins with different chain lengths as the function of temperature is shown in Figure 1b. As the similar results show in Figure 1a, the change of bond-torsional energy per bond as the function of temperature is independent of chain length. Meanwhile, the increasing tendency of the bond-torsional energy is more obvious when the protein at the low-temperature region as the deviation from a straight line. The increased non-bonded energy of each bond with the increasing temperature is shown in Figure 1c, except for the protein at the low temperature. Additionally, the increasing tendency is more obvious when the protein with longer chain as the number of residue–residue pairs is increased with the chain length. The simulation results of the total potential energy including bonded and non-bonded interactions with different chain lengths as the function of temperature is plotted in Figure 1d. The same tendency that the total potential energy is increasing with temperature increases is also presented in Figure 1d. So, we can conclude that the potential energy of the whole chain is only dependent on the chain length when the protein at the certain temperature. In addition, the non-bonded interactions are related to the hydrophobic property of primary sequence of the protein. As the equilibrium state of each protein is mainly due to non-bonded interactions, it will be discussed in detail in the following context.

#### 2.1.2. Radius of Gyration

The radius of gyration, which shows the structural information of a single protein during the simulation process, is mainly discussed. The temperature dependence of the total and components of radius of gyration of a protein chain with the same repeating segment H3N1P1 but different chain lengths are plotted in Figure 2. The curves in Figure 2a show that the *z*-component of radius of gyration $R_{\mathrm{gz}}$ of H3N1P1 protein with chain length N=50 residues is increased rapidly with the temperature decreasing when $T^{*}<0.8$ after the gradual increase in the high temperature region. Meanwhile, $R_{\mathrm{gx}}$(or $R_{\mathrm{gy}}$) keeps constant value or increases slightly with the temperature decreasing, and then has a decreasing trend especially $T^{*}<0.6$. At low temperatures $T^{*}<0.6$, the difference between $R_{\mathrm{gx}}$(or $R_{\mathrm{gy}}$) and $R_{\mathrm{gz}}$ becomes more and more obvious as the temperature reduces, which indicates that the protein chain presents anisotropic configuration. Figure 2b–d show the simulation results on the H3N1P1 proteins with chain length of 120, 240, and 300 beads. The curves of total and components of radius of gyration show the similar trends shown in Figure 2a. Comparing these four figures, for H3N1P1 proteins, the longer the chain length is, the larger the radius of gyration presents. These results indicate that *trans* state is dominated at low temperatures, which makes the ordered structure with obvious anisotropy. Instead, *gauche* states are located everywhere, and a protein takes a random coil configuration when the protein is under the high temperature condition.

#### 2.1.3. Global Orientational Order Parameter

Figure 3 shows the global orientational order parameter *P* of H3N1P1 proteins with the chain length of 50, 120, 240, and 300 residues. It takes a value near 0 under high temperatures, which indicates the protein takes a random coil configuration. After the value of *P* gradually increases with temperature decrease in the range of $T^{*}>0.8$, it increases sharply with decreasing temperature leading to a large growth rate of the orientational order. The increasing order parameter *P* with decreasing temperature indicates the transition of the protein chain from a random coil to an ordered structure. The simulation results on H3N1P1 proteins with different chain lengths show the same tendency. Meanwhile, the shorter chain takes the larger value of *P*, which indicates the bonds are almost paralleled, but the longer protein is partially paralleled under a certain temperature as more continuous bonds influence each other to hardly form a good orientation structure.

We can summarize briefly that the large size of a protein constructed by more residues has the higher non-bonded interactions between residue–residue pairs and low global orientational order indicating the partially paralleled bonds among the protein. The conformational transition from random coil to ordered structure occurs when the temperature drops low enough, and the transition temperature is independent of chain length.

### 2.2. Effect of the HNP Ratios

#### 2.2.1. Energy

In this section, the MD simulation results on the proteins with the same chain length but different residue ratios are mainly discussed. As mentioned above, the non-bonded energy plays a main role to present a specific structure. Then, the temperature dependence of the non-bonded energy of proteins with the different residue ratios are plotted in Figure 4. As the same tendency shown in Figure 1c, the value of the non-bonded energy decreases with the decreasing temperature in all three types of proteins with H4N1P1, H3N1P1, and H4N2P2 residue components. As shown in Figure 4a,b, the higher ratio of hydrophobic residues (H) among the protein chain, the lower non-bonded energy is obtained when the protein is under a certain temperature. The straightforward reason is that only the interaction between H-H residue pairs could be a minus value according to the Equation (4) and parameter values in table in Section 3. When the higher proportion of hydrophobic residues in the repeating unit leads to the larger number of H-H pairs with the attractive interaction between hydrophobic residues, and the non-bonded energy of the whole protein chain is correspondingly lower. It means that the interaction between hydrophobic residues plays an important role in stabilizing the protein structure. Comparing the simulation results in Figure 4a,b, the non-bonded energy decreases more significantly in the case of longer protein N=240 and even presents the minus value under the low temperature when the ratio of hydrophobic residues is high enough.

#### 2.2.2. Radius of Gyration

The curves of the radius of gyration as the function of temperature of H4N1P1, H3N1P1, and H4N2P2 proteins with the chain length of 120 and 240 beads are plotted in Figure 5. All curves show that the radius of gyration of the protein chain increases with the decrease of temperature. This tendency is independent of chain length and repeating unit. But comparing the proteins constructed by different repeating units, for the chain with higher proportion of H residues in a repeating unit may lead to the lower value of the radius of gyration presenting the compact structure comparatively. On the other hand, the structural size of the protein constructed by the low percentage of hydrophobic residues changes more obviously than the one formed by high percentage of H beads. It means the configuration of the protein with low ratio of hydrophobic residues is flexible and easily influenced by temperature. The protein constituted by more hydrophobic residues could form a relatively stable structure that is hardly influenced by temperature. It also demonstrates that more H-H pairs interactions could maintain the stability of the protein structure.

#### 2.2.3. Global Orientational Order Parameter

In order to reveal the effect of HNP proportion on the orientation of the protein chain, the temperature dependence of global orientational order parameter *P* with chain length N=120 and 240 beads but different HNP ratios are plotted in Figure 6. When the protein is in the high temperatures ($T^{*\;}>0.8$) condition, it takes a value near 0, indicating the chain takes a random coil configuration and the numerical difference of *P* among three proportional ratios of HNP residues is slight, especially in the case of longer chain. When $T^{*}<0.8$, *P* increases rapidly with the decreasing temperature, and the higher the H residue proportion in a repeating unit, the higher value of *P*, which indicates a more ordered structure formed. For the longer protein chain, such tendency is more obvious and the numerical difference between protein chains with different HNP proportions is larger than the ones in short protein chains at the low temperature region.

By analyzing the non-bonded energy, the radius of gyration and the global orientational order parameter of the protein, results show that the proteins with the same chain length but different HNP segment proportions in a repeating unit present the different size and orientational characteristics. The one with higher proportion of H residues receives the lower non-bonded energy, the smaller value of radius of gyration, but the larger value of global orientational order, which indicates that the protein chains with a higher proportion of H have more compact and parallel conformation than others.

### 2.3. Effect of the Serial Number of Three Types of Residues with Same Ratio

#### 2.3.1. Energy

The protein chain with the same HNP distribution ratio but different serial residue numbers in the repeating unit are mainly discussed in this part. The non-bonded energies of the protein with different chain lengths as the function of temperature are shown in Figure 7a,b. In these figures, the tendency that the non-bonded energy keeps decreasing with the decreasing temperature is the same in all kinds of proteins regardless of the chain length, residue distribution ratio, and serial number of the same type of residues. For the certain chain length, the protein chain with the higher proportion of H residues in a repeating unit has the lower non-bonded potential energy, which is the same result as mentioned above. On the other hand, the curves with a hollow symbol are always lower than the ones with a solid symbol in any simulation cases. It means that for the protein chains with the same HNP proportion but different serial numbers of hydrophobic, neutral, or hydrophilic residues in the repeating unit, the more H residues in succession along the protein chain can obtain the lower non-bonded energy. It is due to the series of hydrophobic residues forming consecutive contacts easily. Additionally, the negative interaction between hydrophobic residues leads to the lower value of non-bonded energy. Comparing the results presented in Figure 7a,b, although the chain length is doubled, the same tendency that the chain connected by more consecutive H residues has the lower non-bonded energy than the ones with the same ratio but smaller number of consecutive H residues is achieved, especially when the is protein under the low temperature region. We also noticed that in the high temperature region ($T^{*}>0.7$), the non-bonded energy of the H8N4P4 protein is little lower than the ones of the H6N2P2 protein. Here, although the ratio of H residues is lower, the serial number of H residues is larger. It is the synergistic effect of the residue distribution factor and the consecutive number of hydrophobic residues.

#### 2.3.2. Radius of Gyration

The total size of the protein chain constituted by N=120 or 240 residues with the same H, N, and P distribution proportion but different serial residue numbers in the repeating unit is analyzed in this section. The radius of gyrations of each protein chain as the function of temperature are shown in Figure 8. In these two figures, the radius of gyration increases with the decreasing temperature in any simulation cases, which is the same result as in Figure 2 and Figure 5. The protein chain with higher H residue distribution takes the lower value of radius of gyration, which is also obtained in Figure 8. Considering on the effect of serial number of three types of residues with same ratio in a repeating unit, we found that the hollow symbol curves are always lower than the same shape of solid curves. In both Figure 8a,b, the minimum radius of gyration occurs on the protein chain with the repeating unit of H8N2P2, which has a higher H ratio and more sequential hydrophobic residues. It means that for a protein with the same HNP proportion but different repeating units, the one with higher numbers of consecutive H in a repeating unit takes the lower value of radius of gyration because of the compact conformation formed by the mutual attraction between consecutive hydrophobic residues. Briefly, it concluded that the protein chain with higher proportion of H and greater number of consecutive H residues has a tighter configuration.

#### 2.3.3. Global Orientational Order Parameter

The global orientational order parameter *P* of the protein with the same HNP proportion but different serial residue numbers in the repeating unit as a function of temperature is plotted in Figure 9. As the same tendency shown in Figure 3 and Figure 6, *P* increases with the decreasing temperature, which indicates the structural transition from a random coil to an ordered state. For the same chain length, the protein with a larger number of consecutive H residues or a higher proportion of H residues in the repeating unit has the higher value of *P*, indicating the orientationally ordered structure is formed. The longer the chain length is, the more pronounced difference between proteins with different HNP proportions or serial number of same type of residues occurs, especially in the condition of the low temperature region.

Concluding from the above calculation results, for the protein with the higher proportion of H residues and the larger number of serial H residues in a repeating unit, the interaction of hydrophobic residues is more obvious and continuous leading to the lower non-bonded energy, the smaller radius of gyration, and the larger value of *P*. The formation of ordered and compact configuration could be easily obtained in such sequential condition.

#### 2.3.4. Stem Length

In this section, we discuss the conformational change of a protein chain by microscopic analysis. The bond of a protein chain takes the *trans* or *gauche* state, which is defined by the dihedral angle $\left|\varphi \right|\leq \frac{\pi }{3}$ or $\left|\varphi \right|>\frac{\pi }{3}$, respectively. We introduce the concept of a stem defined by more than three consecutive all-*trans* bonds and investigate the stem length distribution at different temperatures. A protein chain consists of several stems with different stem length $N_{\mathrm{tr}}$ (i.e., the number of consecutive *trans* bonds) and connected by loops. The distribution of stem length $N_{\mathrm{tr}}$ was analyzed as the function of temperature.

The distribution of the stem size ${P(N}_{\mathrm{tr}})$ is normalized and shown in Figure 10. An exponential decrease with the increase of stem size is observed at high temperature $T^{*}\geq 1.0$ in all protein samples. The reason for this exponential decrease is that each bond takes the *trans* or *gauche* state randomly with no spatial correlation under the high temperature. It also presents that the rate of exponential decline increases with the increasing temperature especially in the case of H3N1P1 protein in Figure 10a. Besides the exponentially decreased tendency observed in Figure 10b–d, highly non-exponential behavior is also shown when the temperature is low enough. The linear-lg plot of stem length distributions ${P(N}_{\mathrm{tr}})$ of H4N1P1, H6N2P2, and H8N2P2 proteins are deviated from exponential decrease at low temperature $T^{*}\leq 0.5$ and peaks are visible. The unimodal distribution of H4N1P1 protein samples is shown in Figure 10b. Additionally, the peak located at the stem length about 60 bonds becomes more obvious with the temperature decreasing. It indicates that the protein chain is ordered with a specific length at low temperature. Meanwhile, multimodal distributions are shown in the case of H6N2P2 and H8N2P2 protein samples at the low-temperature region in Figure 10c,d. The difference between them is the successive bond number corresponding to the peaks. It means that the protein is ordered with different stem lengths when the proportion of H residues is higher and the number of serial H residues is larger in a repeating unit. Then, both the proportion and the consecutive number of H residues in a repeating unit would affect the stem length of the ordered protein. According to the simulation results, we may infer that not only the H-occupancy ration but also the number of consecutive hydrophobic residues affect the stem length and its distribution. With the number of consecutive H increases, there will be a transition from no peak to single peak and even multiple peaks. Most of the stems are fixed to one or several specific lengths, indicating the ordered protein chain with specific lengths under the low temperature.

#### 2.3.5. Contacts Number

Two residues that are close in space and far in the sequence are often defined as contact to describe the effect of long-range interactions on the structural characteristics of the protein. The number of contacts on the structural level of protein samples with different ratios and sequential hydrophobic residues were statistically analyzed. Nonlocal residue–residue contacts are defined as two $C^{\alpha }$ atoms within a cutoff distance $R_{cut}=0.65\;\mathrm{nm}$ and separated by at least a residue separation cutoff value $l_{cut}=3.$ The statistical results are plotted in Figure 11. It presents that the contacts number of the protein H4N2P2 and H3N1P1, which is constituted by the low ratio and small sequential number of hydrophobic residues, is independent of the temperature. For the other protein chains in Figure 11, contacts number is independent of the temperature when the protein in the high temperature range $T^{*}\geq 1.0$. With the temperature decreasing, more contacts formed according to the lower potential energy between $C^{\alpha }$ atoms. It is the similar tendency with the global orientational order parameter as the function of temperature in Figure 9b. More contacts between residues could make the chain orientated to the ordered configuration. In Figure 11, it also shows clearly that proteins made up with a higher constituent ratio of hydrophobic residues are easy to form contacts. Meanwhile, comparing the protein chains with the same ratio but different repeat units, the protein with large sequential number of hydrophobic residues are easy to form contacts. Here, the discussion of contact number could also evaluate the long-range interactions among the protein and present the connection of its structure.

## 3. Materials and Methods

To minimize the running time during simulation processes and generalize the general properties of proteins instead of a specific one, the modified off-lattice HNP model is applied in this study [21]. Twenty types of residues are classified into three groups, namely hydrophobic residues (H) consisting of Phe, Met, Ile, Leu, Trp, Val, Cys, and Tyr, neutral residues (N) consisting of His, Ala, Gly, and Thr, and hydrophilic residues (P) consisting of Lys, Asp, Asn, Glu, Gln, Ser, Pro, and Arg [22]. Then, a single protein chain is described by a sequence of three types of beads, H, N, and P. Each bead in a protein molecule is represented by the position of a $C^{\alpha }$ atom. The beads interact via the non-bonded potential and bonded potentials, which including bond-stretching, bond-bending, and bond-torsional potentials. The potential energy expressions and the parameter values of the coarse-grained model are adopted as in the literature [21,23,24]. It can be thought of a simple version used in the study of the structure and dynamics of real proteins. The behavior of a single protein chain with different distributions of H, N, and P beads has been studied by means of MD simulations here. The four potential functions of a protein are as follows:(I)The bond-stretching potential energies for every couple of adjacent beads connected by covalent bonds,
(1)$$E_{\mathrm{stretch}\;}=\overset{N}{\underset{i=2}{\sum }}\frac{1}{2}k_{d}{{(l}_{i}-l_{0})}^{2}$$
where the equilibrium bond length *l*_0_ = 0.153 nm and *l_i_* is the bond length between beads *i* − 1 and *i*. The bond-stretching potential energy constant *k_d_* = 7 × 10^4^ kcal/nm^2^mol.
(II)The bond-bending potential energies are defined for every triplet of adjacent beads,
(2)$$E_{\mathrm{bend}\;}=\overset{N}{\underset{i=3}{\sum }}\frac{1}{2}k_{\theta }{\left(\theta_{i}-\theta_{0}\right)}^{2}$$
where the equilibrium bond angle *θ*_0_ = 1.231 rad and *θ_i_* is the bond angle between three beads *i* − 2, *i* − 1, and *i*. The bond-bending potential energy constant *k_θ_* = 100 kcal/rad^2^mol.
(III)The bond-torsional potential energies are defined for every quadruplet of adjacent beads,
(3)$$E_{\mathrm{torsion}}=\overset{N}{\underset{i=4}{\sum }}\frac{1}{2}k_{\varphi }\left(1+\cos\left({3\varphi }_{i}\right)\right)$$
where *φ_i_* is the dihedral angle of two planes consisting of four consecutive beads *i* − 3, *i* − 2, *i* − 1, and *i* along the backbone. The constant of torsional energy *k_φ_* = 2.0 kcal/mol. The threefold torsional energy of a bond in a polymer generally exhibits several stable states: *trans* (t), *gauche* plus (g^+^), and *gauche* minus (g^−^) separated by barriers. The continuous *trans* states in the polymer chain show the rigidity of chain molecules with parallel ordering configuration.
(IV)The non-bonded interaction involves beads that are at least four residues apart in the amino acid sequence. For two beads, the non-bonded interaction is given by the truncated 12-6 Lennard–Jones potential,
(4)$$E_{\mathrm{LJ}}=\overset{N}{\underset{i=1}{\sum }}\overset{N}{\underset{j=1}{\sum }}4\varepsilon \left[{\left(\frac{\sigma }{R_{\mathrm{ij}}}\right)}^{12}-\Lambda {\left(\frac{\sigma }{R_{\mathrm{ij}}}\right)}^{6}\right]\;\left(j\;-\;i\geq 4\right)$$
where *r_ij_* is the distance between atoms *i* and *j*. ε is the parameter that defines the energy strength and Λ sets the nonlocal interaction that is attractive or repulsive. Both of these two parameters depend on the nature of the two residues involved. The values of the van der Waals interaction parameters are shown in Table 1. Here, the values of potential parameters are $\varepsilon_{h}$ = 0.1984 kcal/mol and σ = 0.3624 nm.

The mass of protein chain is fixed to cancel overall chain translation. The coordinate system with the three principal axes of inertia is introduced to analyze the anisotropic structure. The radius of gyration $R_{g}$ is a basic measurement of the overall size of a protein chain. The change in the structure of a protein during MD simulations can be quantified by the radius of gyration $R_{g}$, which is defined as:(5)$$R_{g}=\sqrt{\frac{1}{N}\overset{N}{\underset{i=1}{\sum }}|r\left(i\right)-r_{\mathrm{center}}{|}^{2}}$$
where *N* is the number of protein atoms. *r*(*i*) and *r_center_* are the coordinates of a bead *i* and the center of mass, respectively. For in-depth understanding of the average shape of a protein chain, a three inertial principal axis coordinate system is introduced, and the relation $R_{g}^{2}=R_{gx}^{2}+R_{gy}^{2}+R_{gz}^{2}$ holds. We also calculate the bond-orientational order parameter *P* defined by
(6)$$\;P=\frac{1}{N-2}\overset{N}{\underset{i=3}{\sum }}(\frac{{3\cos²\varphi }_{i}-1}{2})$$
where *φ_i_* is the angle between the bond vector and the *z*-axis of the protein chain coordinate system and *N* is the number of beads. The parameter *P* would take a value of 1 for a protein whose bonds are perfectly parallel, and that of 0 for randomly oriented. We have monitored the value of *P* for the whole chain.

The same as the previous work, all simulation results were presented in terms of reduced units [25]. It follows that the reduced temperature $T^{*}\;$ is equal to $k_{B}T/\varepsilon$, the reduced energy is E/ε, and the reduced time is $t\sqrt{{\varepsilon /m\sigma }^{2}}$. A single protein chain consisting of three types of beads with different chain lengths is exposed to a vacuum. The MD simulations are carried out from random configuration of the protein at high temperature. Then, the protein is quenched to a certain low temperature, and 500 ns (1 × 10^7^ time steps) simulations are carried out under several temperature conditions, which takes $T^{*}$ from 0.3 to 1.5. The equations of motion for all atoms are solved numerically using the velocity version of the Velocity–Verlet algorithm [26]. The Nosé–Hoover method is applied to keep the system temperature constant [27,28]. The integration time step is 2.5 fs and a relaxation constant is 25 fs, respectively. The cutoff distance is 0.95 nm in our simulation. The total linear and angular momentums are taken to be zero in order to cancel overall translation and rotation of the chain.

Based on the idea of the HNP model, proteins are described as a linear polymer consisting of three beads with different properties as basic units. For example, H3N1P1 can be described as a protein chain with three hydrophobic residues, one neutral residue, and one hydrophilic residue connected as a repeat unit. According to this model, we investigate the effects of the composition and distribution of amino acid in protein primary sequence on the 3D structure of the protein through calculating the energies and structural parameters. The potential and structural properties of the protein chain with different lengths and HNP components in terms of both energy and structural parameters have been discussed in this paper.

## 4. Conclusions

In this article, we performed MD simulations for the protein chain with different HNP segment proportions and the same HNP proportion but different repeating units over a wide range of temperatures. We investigated the potential energy and structural parameters such as radius of gyration, global orientational order parameter, stem length distribution, and contact number of the protein. The current results indicate:

(1)The four potential energy of each bond in the protein chain is independent of chain length. The linear relationship between the bond-stretching/bending energy per bond and the temperature is obtained. It is consistent as the equipartition theorem of energy that the average value of bond-stretching/bending energy associated with each independence degree of freedom bond length *l* or bond angle *θ*. The increasing tendency of the bond-torsional or non-bonded energy has a little deviation from the straight line. Overall, the total potential energy of the whole chain is only dependent on the chain length when the protein is at a certain temperature.The total and components of the radius of gyration with different protein chain lengths as the function of temperature show that $R_{g}$ increases with decreased temperature. $R_{\mathrm{gz}}$ shows a similar tendency with $R_{g}$, while $R_{\mathrm{gx}}$(or $R_{\mathrm{gy}}$) presents the opposite tendency with $R_{g}$, especially when $T^{*}<0.6$. The differences between $R_{g}{/R}_{\mathrm{gz}}$ and $R_{\mathrm{gx}}{/R}_{\mathrm{gy}}$ indicate that the protein takes anisotropic configuration at the low temperature. The increasing global orientational order parameter *P* with the decreasing temperature indicates the paralleled bonds among the protein. The shorter chain takes the larger value of *P* than the one of longer protein, as the longer protein is partially paralleled under a certain temperature with more continuous bonds influencing each other to hardly form a good orientated structure.(3)The effect of the hydrophobic, neutral, and polar residue proportions under certain chain length was also investigated. The simulation results show that the protein with the higher proportion of hydrophobic residue in a repeating unit takes the lower value of non-bonded energy, the lower value of radius of gyration and the higher value of global orientational order when protein chains with the same length under the certain temperature. It indicates that the protein with higher ratio of hydrophobic residues can easily transform from a random coil to an oriented and compact structure with the decreasing temperature.(4)From the perspective of the successive number of hydrophobic residues in a repeating unit, we concluded that the protein with a higher number of consecutive H residue receives the lower value of the non-bonded energy, the lower value of the radius of gyration and the larger value of global orientational order as the sequential hydrophobic residues make more consecutive H-H interaction pairs to form tight and ordered configuration. The analysis of contact number also reflects the long-range interaction among the residues.(5)Exponential decline of stem length distribution is shown in any proteins under the high temperature situation. However, with the temperature decreasing, different proteins present a diverse tendency of stem length distribution. For example, the stem length distribution of H3N1P1 protein chains can keep exponentially decreased in the low temperature, while H4N1P1, H6N2P2, and H8N2P2 present peaks located at different stem lengths from 30 to 60 bonds. It indicates the specific ordered lengths of the ordered protein chain. The protein with higher proportion or larger number of consecutive hydrophobic residue H in a repeating unit present the transition of the stem length distribution from exponential decline to unimodal peak and even multiple peaks Thus, the proportion and the number of consecutive hydrophobic residue influence the stem length distribution simultaneously.

## Figures and Tables

**Figure 1 ijms-23-14263-f001:**
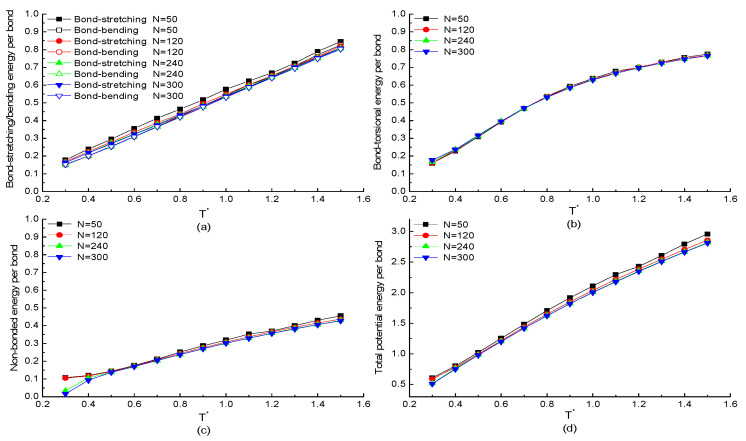
The temperature dependencies of the potential energies (**a**) bond-stretching/bending energy, (**b**) bond-torsional energy, (**c**) non-bonded energy, (**d**) total potential energy of each bond in H3N1P1 proteins with different chain lengths.

**Figure 2 ijms-23-14263-f002:**
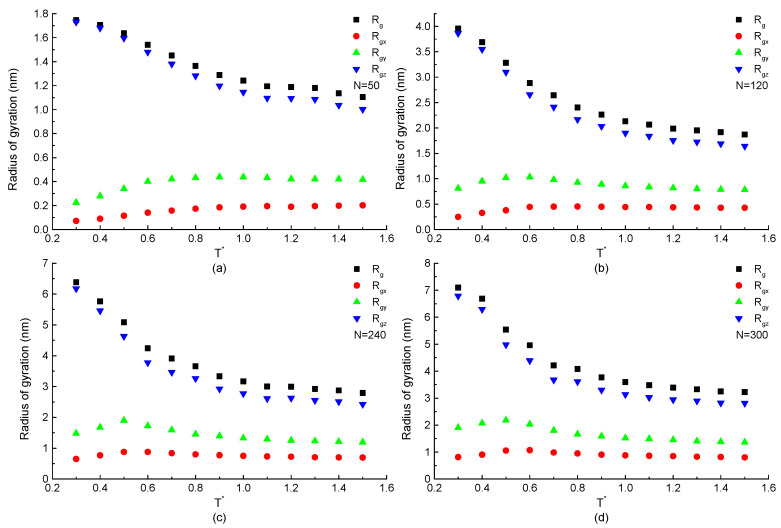
The total and components of radius of gyration versus reduced temperature $T^{*}$ of H3N1P1 proteins with chain length (**a**) N=50, (**b**) N=120, (**c**) N=240, and (**d**) N=300 residues.

**Figure 3 ijms-23-14263-f003:**
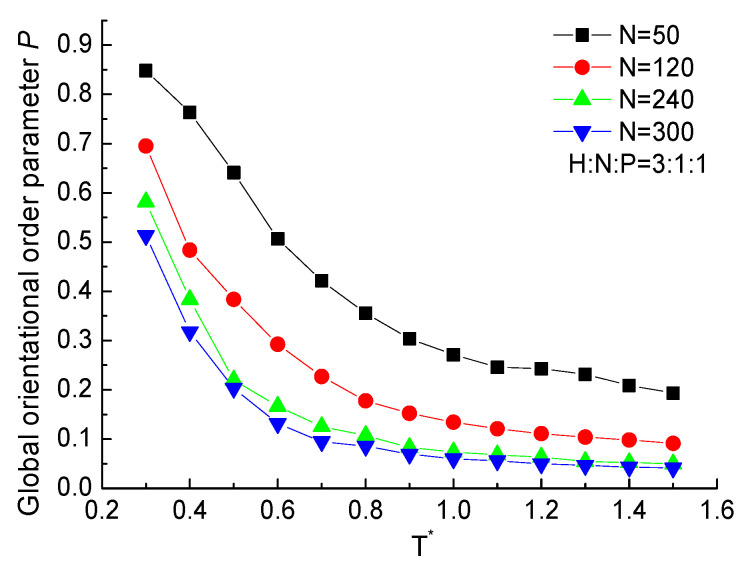
The global orientational order parameter *P* versus reduced temperature $T^{*}$ of H3N1P1 proteins with different chain lengths.

**Figure 4 ijms-23-14263-f004:**
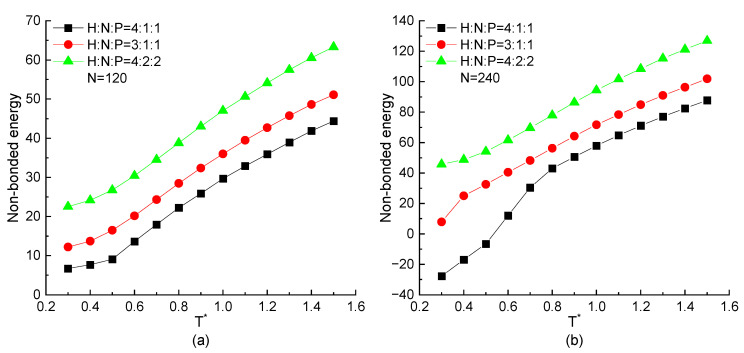
The non-bonded energy versus reduced temperature $T^{*}$ of H4N1P1, H3N1P1, and H4N2P2 proteins with chain length of (**a**) N=120, (**b**) N=240.

**Figure 5 ijms-23-14263-f005:**
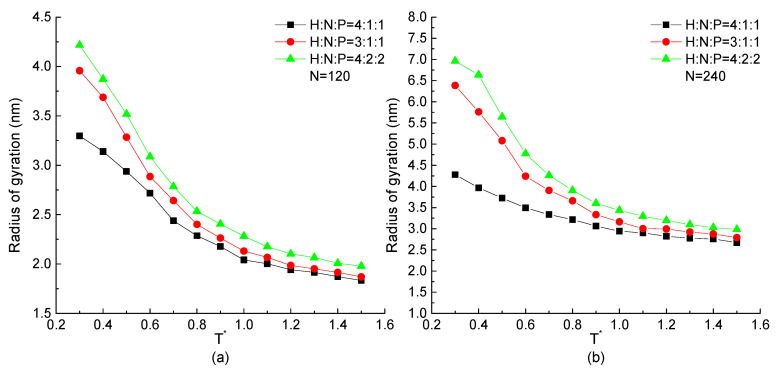
The radius of gyration versus reduced temperature $T^{*}$ of H4N1P1, H3N1P1, and H4N2P2 with chain length of (**a**) N=120, (**b**) N=240.

**Figure 6 ijms-23-14263-f006:**
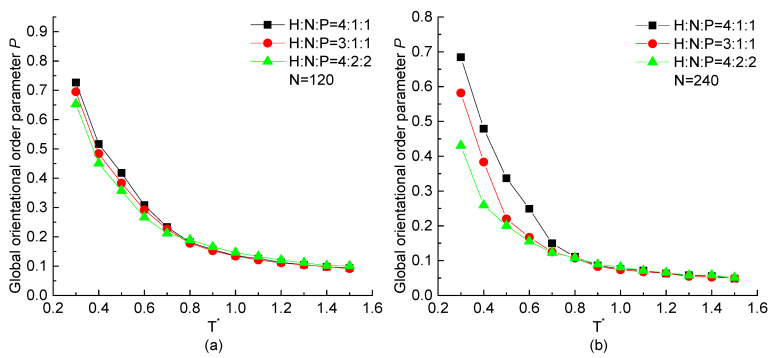
The global orientational order parameter *P* versus reduced temperature $T^{*}$ of H4N1P1, H3N1P1, and H4N2P2 with chain length of (**a**) N=120, (**b**) N=240.

**Figure 7 ijms-23-14263-f007:**
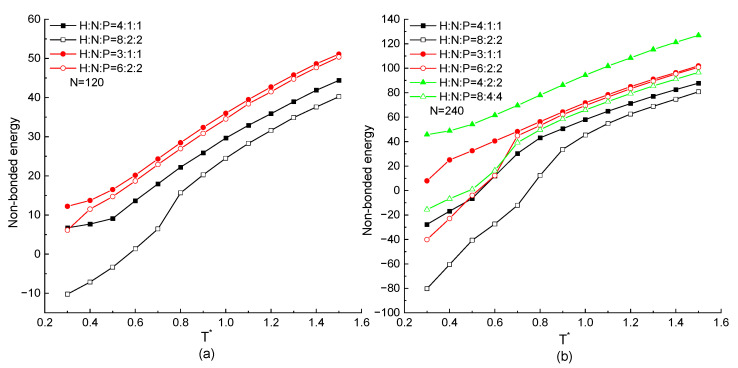
The non-bonded energy versus reduced temperature $T^{*}$ of different ratio and serial number of H, N, and P residues with protein chain length (**a**) N=120, (**b**) N=240.

**Figure 8 ijms-23-14263-f008:**
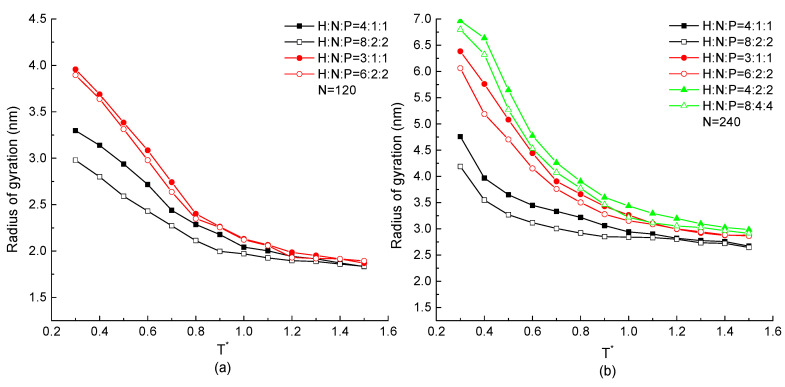
The radius of gyration versus reduced temperature $T^{*}$ of different ratio and serial number of H, N, and P residues with protein chain length (**a**) N=120, (**b**) N=240.

**Figure 9 ijms-23-14263-f009:**
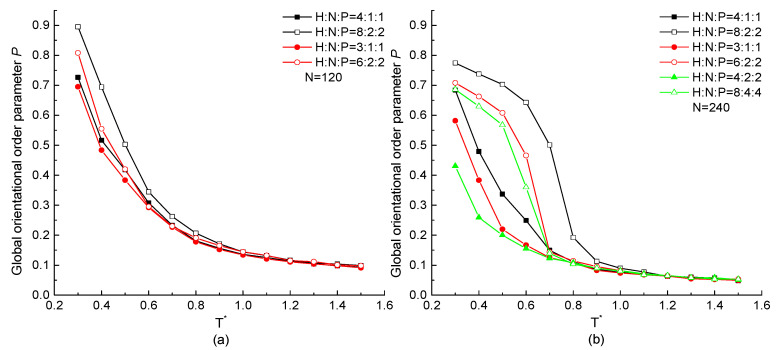
The global orientational order parameter *P* versus reduced temperature $T^{*}$ of different ratio and serial number of H, N, and P residues with protein chain length (**a**) N=120, (**b**) N=240.

**Figure 10 ijms-23-14263-f010:**
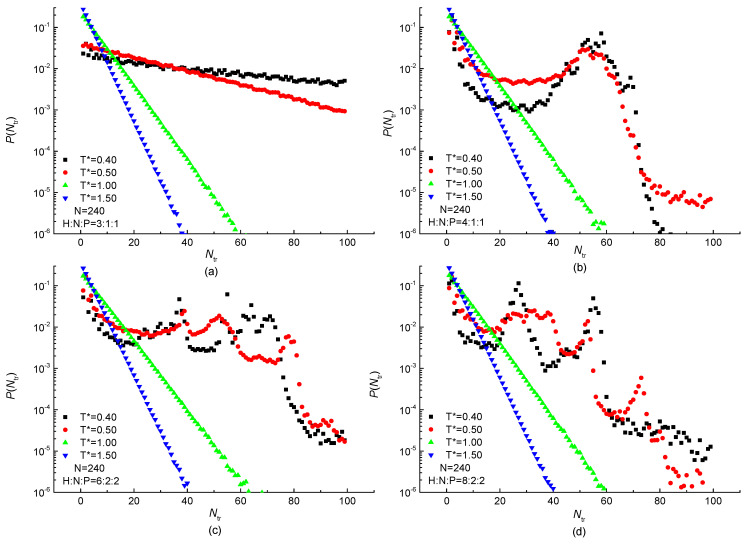
Distribution of stem length under different temperatures in linear-lg plot of (**a**) H3N1P1, (**b**) H4N1P1; (**c**) H6N2P2 and (**d**) H8N2P2 proteins with chain length of N=240 residues.

**Figure 11 ijms-23-14263-f011:**
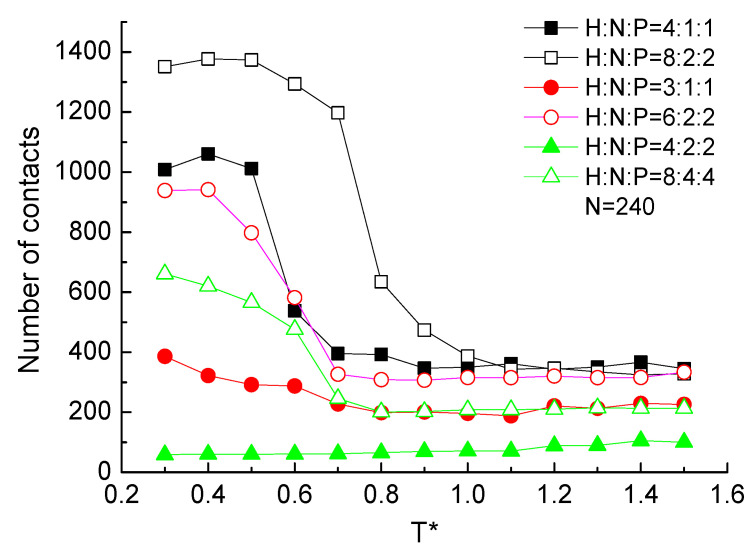
The number of contacts versus reduced temperature $T^{*}$ of different ratio and serial number of H, N, and P residues with protein chain length N=240.

**Table 1 ijms-23-14263-t001:** Parameters ε and Λ in the non-bonded interaction term of the potential function depending on the nature of the two residues involved.

	H	N	P
H	$\varepsilon =\varepsilon_{h};\;\Lambda =1$	$\varepsilon =\frac{2}{3}\varepsilon_{h};\;\Lambda =0$	$\varepsilon =\frac{7}{12}\varepsilon_{h};\;\Lambda =0$
N	$\;\varepsilon =\frac{2}{3}\varepsilon_{h};\;\Lambda =0$	$\varepsilon =\frac{1}{3}\varepsilon_{h};\;\Lambda =0$	$\varepsilon =\frac{1}{4}\varepsilon_{h};\;\Lambda =0$
P	$\;\varepsilon =\frac{7}{12}\varepsilon_{h};\;\Lambda =0$	$\varepsilon =\frac{1}{4}\varepsilon_{h};\;\Lambda =0$	$\varepsilon =\frac{1}{6}\varepsilon_{h};\;\Lambda =-1$

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
