# Peer review of "Effects of Residual Composition and Distribution on the Structural Characteristics of the Protein"

_ijms, 2022, doi:10.3390/ijms232214263_

Round 1
Reviewer 1 Report
The authors presented a study using molecular simulation to investigate the protein backbone folding based on an off-lattice coarse-grain model. The model of interest classifies amino acid residues into 3 categories, namely hydrophobic (H), hydrophilic (P) and neutral (N), which defines the interaction term among each other. No other long-range interaction was taken into consideration, including hydrogen bond. The effect of temperature, protein length and composition on the protein folding were studied.
Although the use of coarse-grain model is widely adopted in polymer and protein MD study, the negligence of hydrogen bond is questionable. The hydrogen bond is the key component responsible for the secondary structure of protein, such as a-helix and b-sheet. It is quite obvious in this study that the absence of hydrogen bonding contribution leads to folding of the protein instead of unfolding when temperature increases. For example, Fig.2 suggested that the radius of gyration Rg decreased with respect to increase of temperature, which corresponds to an unrealistic “fold” upon heating. The same observation is made in Fig.3, where the protein backbone is essentially straight (P~1) at low temperature, and more “folded” (P<<1) at high temperature. I would strongly suggest the authors to include hydrogen bonding interactions in their model and at least reproduce the unfolding process of protein during heating.
There are other details worth revisiting in this study. For example in Fig.1, the bonded energies were normalized by chain length, but the non-bonded terms were not. Then they reached the conclusion that the non-bonded energies scaled with temperature, but not the protein-length-normalized bonded energies. I’m confused by this conclusion, as the total energy should be normalized by protein length, hence the non-bonded terms contribute to it.
I hope the authors could revisit this study and address the questions above.
Reviewer 2 Report
· The article has an intense mathematical background. So check the notations and proofs
· The novelty is not clear, and its relevance to the journal's scope needs highlighted in abstract, introduction and throughout.
· What is the innovation of this paper? The methods introduced in this paper are all existing methods.
· There are some grammatical problems in the text, and the description of this paper needs further improvement.
· In the introduction part, the logic is confused, the author should reorganize. Meanwhile, the author should give the existing problems solved by the proposed method, and what are the contributions and innovations of this paper? Modeling different structures in perturbed Poiseuille flow in a nanochannel by using of molecular dynamics simulation: Study the equilibrium Molecular dynamics simulation of fluid flow passing through a nanochannel: effects of geometric shape of roughnesses Prediction of boiling flow characteristics in rough and smooth microchannels using molecular dynamics simulation: Investigation the effects of boundary wall temperatures Investigation of thermal properties of DNA structure with precise atomic arrangement via equilibrium and non-equilibrium molecular dynamics approaches Molecular dynamics simulation of Couette and Poiseuille Water-Copper nanofluid flows in rough and smooth nanochannels with different roughness configurations
· Please edit the paper carefully such that to respect the instructions for authors. A homogeneous style is desired.
· The introduction section is too long. Please focus the idea of the introduction section and not writing as an exposition of different essays.
· English grammar must be checked and corrected thorough the manuscript.
·
· You should present the contributions with respect to your past papers that should be cited. Your past algorithms are very well appreciated.
· References that do not belong to the main stream publications should be deleted.
· I am not sure if the comparison is correct because all algorithms and classifiers used in the comparison including yours depend on parameters. Other parameters lead to other results
· In the conclusions section, an explanation can be made about the reasons why the results obtained as a result of the study are not 100% accurate. Namely; Is this due to a lack of filtering?
Round 2
Reviewer 1 Report
I agree with the author's argument that the goal of course grain model is to simplify the simulation, but it does not qualify a model that did not represent a basic feature of the object, which is a protein would unfold (higher entropy and larger Rg) at high temperature. Even the additional figure included in the response showed that the Rg decreases as T increases. It's unclear to me how informative to discuss Rg out of the content of solvent.
What I can see in this model is that a well-ordered extended 1D crystalline-like structure is analogous to "folded" structure of protein, and the deviation from it is considered unstructured and "unfolded". However, it is not the common practice in the field, please avoid "folded" and "unfolded" and probably use "structured" vs "unstructured".
Regarding the normalization of the total energy, my comment was about that Fig.1d seemingly indicate that the total energy is almost linear to chain length, which might as well be normalized to the chain length for the sake of clarity, therefore normalizing both non-bond and total energy.
It would be really helpful to reader like me if the terminology and concepts in the manuscript were to clarified.
Round 3
Reviewer 1 Report
The concepts and aims are now much better delivered after the modifications.